# MicroRNA Dysregulation in Early Breast Cancer Diagnosis: A Systematic Review and Meta-Analysis

**DOI:** 10.3390/ijms24098270

**Published:** 2023-05-05

**Authors:** Alejandro Garrido-Palacios, Ana María Rojas Carvajal, Ana María Núñez-Negrillo, Jonathan Cortés-Martín, Juan Carlos Sánchez-García, María José Aguilar-Cordero

**Affiliations:** 1CTS367, Andalusian Plan for Research, Development and Innovation, University of Granada, 18001 Granada, Spain; 2Department of Nursing, Faculty of Health Science, University of Granada, 18001 Granada, Spain; 3CTS1068, Andalusian Plan for Research, Development and Innovation, University of Granada, 18001 Granada, Spain

**Keywords:** micro-RNA, cancer, biomarkers, miRNA diagnostics, breast cancer

## Abstract

Breast cancer continues to be the leading cause of death in women worldwide. Mammography, which is the current gold standard technique used to diagnose it, presents strong limitations in early ages where breast cancer is much more aggressive and fatal. MiRNAs present in numerous body fluids might represent a new line of research in breast cancer biomarkers, especially oncomiRNAs, known to play an important role in the suppression and development of neoplasms. The aim of this systematic review and meta-analysis was to evaluate dysregulated miRNA biomarkers and their diagnostic accuracy in breast cancer. Two independent researchers reviewed the included studies according to the preferred reporting items for systematic reviews and meta-analyses (PRISMA) guidelines. A protocol for this review was registered in PROSPERO with the registration number “CRD42021256338”. Observational case-control-based studies analyzing concentrations of microRNAs which have been published within the last 10 years were selected, and the concentrations of miRNAs in women with breast cancer and healthy controls were analyzed. Random-effects meta-analyses of miR-155 were performed on the studies which provided enough data to calculate diagnostic odds ratios. We determined that 34 microRNAs were substantially dysregulated and could be considered biomarkers of breast cancer. Individually, miR-155 provided better diagnostic results than mammography on average. However, when several miRNAs are used to screen, forming a panel, sensitivity and specificity rates improve, and they can be associated with classic biomarkers such us CA-125 or CEA. Based on the results of our meta-analysis, miR-155 might be a promising diagnostic biomarker for this patient population.

## 1. Introduction

The incidence of breast cancer follows a growing pattern, especially in recent decades. Based on data from the GLOBOCAN 2020 study, this neoplasm exceeded the incidence of lung cancer, establishing it as the most prevalent neoplasm worldwide [1]. Regarding mortality, breast cancer constitutes the second oncological cause of death in women. Accordingly, it is estimated that one in four women will suffer from breast cancer throughout her life and one in six will die from this ailment.

Early diagnosis notably improves the long-term survivability rates of breast cancer [2]. The most widely used diagnostic techniques for breast cancer screening consist of imaging techniques. Among them, mammography is considered the gold standard technique in breast cancer screening. Similarly, ultrasounds have shown great efficacy as a complementary technique to detect potential false-negative cases [3]. Other advanced imaging techniques, such as magnetic resonance imaging (MRI) or positron emission tomography (PET), have notable specificity and sensitivity values, enabling the identification of lesions that cannot be detected by other means, albeit their high costs hamper their routine use as early-stage breast cancer detection screening tools [4,5]. 

Mammography has severe limitations in ages outside the range of 40–59 years. In this sense, this screening method is especially inaccurate in patients below 40 years of age, causing underdiagnosis [6]. Furthermore, the incidence of triple-negative tumors in this age range is higher, which implies worst prognostics. Owing to the idiosyncrasy of this age range, cancers such as pregnancy-associated breast cancer (PABC), for which the mortality rate is around 50–60%, are particularly prevalent [7,8,9].

The overall cancer survival rate notably improves if a patient is diagnosed prior to the occurrence of distant metastasis [10]. Analysis of biomarkers might overcome the limitations of imaging techniques whilst enabling the early diagnosis of the disease. In this regard, the human epidermal growth factor receptor 2 (HER2), the KI-67 protein, and estrogen receptors (ERs) are typically used for prognosis and guidance regarding systemic treatment. Among other potential biomarkers of breast cancer are miRNAs, which have gained momentum in recent years. miRNAs are non-coding small RNA molecules of 21–25 nucleotides that mediate the downregulation of target proteins [11]. The human genome is 3 × 10^9^ base pairs long. On the human genome, we currently have around 2000 miRNAs annotated, but each (mature) miRNA is just 20–25 base pairs long, thus representing a tiny fraction of the human genome in terms of sequence length [12]. MiRNAs can be involved in the suppression or activation of tumors. Neoplasm-activating miRNAs are called “onco-miRNAs”.

Circulating miRNAs are regulators of gene expression and mediators of intercellular communication. They are also perfect candidates for a new class of non-invasive biomarkers for the diagnosis, prognosis, and therapeutic evaluation of cancer. There are several justifications this approach; miRNAs present in plasma and serum have high stability, making blood collection reproducible and non-invasive. At the same time, the deregulation of miRNA expression has been associated with cancer. Notable miRNA expression profiles in blood appear to be tissue-specific: miR-122 is preferentially expressed in the liver and miR-133 in muscle. Circulating levels of miRNAs are known to return to baseline levels after tumour removal, which justifies the potential usefulness of circulating miRNAs as biomarkers of cancer treatment efficacy [11].

The deregulation of onco-miRNAs has been widely studied in serum, relating its dysregulation to the existence of an incipient or established neoplasm.

We aimed to systematically evaluate miRNAs with diagnostic potential in breast cancer. Subsequently, we conducted a diagnostic test accuracy meta-analysis to evaluate the diagnostic potential of the most cited miRNA in the systematic review, miRNA-155.

## 2. Materials and Methods

To achieve the proposed objectives, we conducted a systematic review following the Preferred Reporting Items for Systematic Reviews and Meta-Analysis (PRISMA) statement. We conducted a qualitative analysis to screen microRNAs with diagnostic potential. Subsequently, a quantitative data synthesis was conducted to evaluate the diagnostic accuracy of specific miRNAs. This review was registered in the International Prospective Register of Systematic Reviews (PROSPERO) with the registration number “CRD42021256338”.

### 2.1. Search Strategy

A thorough search of relevant articles from the PubMed (Medline), Scopus, and CINAHL databases was performed by one reviewer. Considering that miRNA concentrations in individuals with and without breast cancer have been extensively reported in previous decades, the studies were limited only to those published between 2010 and 2021, using different combinations of the following search terms: “microRNA”, “miRNA”, “miR”, “breast cancer”, “breast neoplasm”, “early diagnosis”, and “screening”. The searches were limited to articles published in English. Keywords were selected using the Medical Subject Heading (MeSH) terms based on the patients, interventions, comparators, results, and study design defined for the present study (PICO). The results obtained through the different search queries were combined using the Mendeley software (v.1.19.8), eliminating duplicated articles. Two reviewers screened the titles and abstracts of studies identified in the initial search to determine the relevance of these publications. Then, full texts were obtained and reviewed in detail. Finally, 38 articles were included in this systematic review after the screening and selection processes.

### 2.2. Study Selection

Articles were considered eligible if they met all of the following inclusion and exclusion criteria:

The inclusion criteria consisted of: (1) studies conducted on pregnant women, including cases of breast cancer diagnosed by a validated method; (2) case-control studies; (3) studies assessing the expression of breast cancer-related miRNAs; (4) sample sizes ≥ 20.

The exclusion criteria included: (1) studies conducted on animal models; (2) studies with very high risk of bias, evaluated using the NOS scale (Newcastle-Ottawa Scale for Observational Studies), which is detailed in Appendix A; (3) studies that use an miRNA for normalization with poor scientific evidence of its variation in cases versus controls; (4) studies including women with other diagnosed pathologies that might bias results (i.e., another type of cancer). 

The meta-analysis inclusion criteria were as follows: (1) miRNAs evaluated in least three different studies.; (2) articles reporting sufficient raw data to construct a 2 × 2 contingency table of true positives (TP), true negatives (TN), false positives (FP), and false negatives (FN), or providing sensitivity and specificity values. 

The exclusion criteria included only one item: low quality assessed by the QUADAS-2 tool.

### 2.3. Quality Assessment

Two independent reviewers assessed the risk of bias in the selected studies using the Newcastle-Ottawa scale for observational studies of case-control design. Studies with a score of 0–3 stars were considered to present a very high risk of bias and were thus excluded from this review. Studies with a score of 4–6 were considered to present a high risk of bias, and studies that achieved a score of ≥7 were considered to be of a high quality/present a low risk of bias.

To ensure the quality of the meta-analysis, the six studies included were evaluated by following the second version of the Quality Assessment of Diagnostic Accuracy Test (QUADAS-2), which evaluates four key domains: patient selection, index test, reference standard, and flow and timing. Studies with a score of less than 7 were considered to have a high risk of bias and low applicability and were thus excluded from this meta-analysis. Disagreements regarding individual studies were discussed and resolved to reach a consensus between the reviewers. 

### 2.4. Data Extraction

A template was created with the following data: author, year, age, normalization method, sample size of all groups, miRNA(s) profiled, and sample source. The extracted data were managed using the software Microsoft Excel^®^. Data were extracted independently by two reviewers. Whenever disagreements between the reviewers occurred, a third reviewer was the arbiter.

### 2.5. Statistical Analysis

For diagnostic accuracy, the included studies’ sensitivity, specificity, diagnostic odds ratio (DOR), positive and negative likelihood ratios (PLR and NLR), and corresponding 95% CI were pooled to assess the diagnostic value of miRNA-155 in breast cancer. The summary receiver operating characteristic (SROC) curve was plotted based on the original data, and the area under the SROC curve (AUC) was calculated to determine the diagnostic accuracy of miR-155. The I^2^ and Chi tests were conducted to estimate the proportion of total variation among studies that occurred due to heterogeneity rather than chance. I^2^ values of >50% were considered to be indicative of significant heterogeneity among the included studies, and thus a random effects model was applied in the analyses [13]. Quantitative analysis of publication bias was assessed using the Deek’s test and creating funnel plots. Subsequently, meta-regression analyses were carried out to find the potential sources of heterogeneity. Finally, the risk of publication bias of all the included studies was estimated by Deeks’ funnel plots [14]. All statistical analyses were performed using RevMan and Stata version 12 (Stata Corporation, College Station, TX, USA).

## 3. Results

### 3.1. Characteristics of the Included Studies

The screening process of the eligible studies is presented in Figure 1. To study the diagnostic value of miRNAs, a total of thirty-eight articles were finally included for qualitative synthesis. Out of the total selected original studies, thirty examined miRNAs’ expression levels between confirmed cases and healthy controls. The remaining eight studies compared the expression of selected miRNAs between a cancerous tissue versus an adjacent healthy tissue. The data extracted from the included papers are presented in Table 1. In Table 2, data on specific miRNAs were drawn and pooled from the included studies according to the combined sample of all those studies in which they were analyzed, including total cases and controls, how they were found to be dysregulated, the biological sample in which they were measured, and their relative value as reported by the authors.

### 3.2. RNA Quantification

All candidate-driven studies used RT-qPCR (real-time quantitative polymerase chain reaction) to detect and quantify miRNA levels. Several studies used in screening-phase microarrays or low-density PCR arrays. Subsequent validation of candidate miRNA biomarkers was performed with RT-qPCR [15,20,21,23,27,34,37,39,42,45].

### 3.3. Data Normalization

In most studies, some form of normalization of circulating miRNA levels was reported to help compensate for potential variations between biological species. miR-16 was used as endogenous control in most of the studies (40%) [15,16,20,21,23,25,27,30,31,34,37,39,42], following by miR-39 and RNU38B. Articles that used an endogenous control that had been reported to be poorly detectable and highly variable were excluded.

### 3.4. Quality of Included Studies

The evaluation of the methodological quality of the studies analyzed in this review is shown in the Appendix A. Thirty-four studies had results that ranged from seven to nine stars (high quality studies), and four articles had between five and six stars. All of the studies selected cases according to valid diagnostic tests (i.e., biopsy) or independent validation, excluding cases for whom breast cancer was not primary. The controls belonged to the same community (generally the same hospital from which the cases were recruited) and had no history of disease.

All of the studies included scored more than seven out of ten. As a result, significant bias was not presented in the meta-analyses, as suggested in Figure 2, which represents detailed information regarding QUADAS-2 assessment. One article was excluded for high risk of bias, and four were excluded for missing information (see Figure 2 for detailed QUADAS-2 score).

### 3.5. Differentially Expressed miRNAs

In the studies comparing breast cancer cases and healthy controls, 34 miRNA were dysregulated. the most cited miRNA was miR-21 [15,24,25,26,27,28,29,30,34,36,38,39,42,49,50,51], followed by miR-155 [18,19,20,23,24,25,33,36,37,39,44,46,47,48,49].

Two miRNAs (miR-10b and miR-16) showed no significative dysregulation, indicating their value as endogenous controls.

### 3.6. Diagnosis Test Accuracy Meta-Analysis

Six articles fit the inclusion and exclusion criteria to be part of this meta-analysis. Figure 3 indicates the individual characteristic of the articles included.

#### 3.6.1. Pooled Diagnostic Value of miR-155 in Breast Cancer

The forest plots of sensitivity and specificity are presented in Figure 4 as follows: the pooled sensitivity and specificity were 86% (95% CI: 70–94%) and 93% (95% CI: 79–98%), respectively. The PLR and NLR were 11.53 (95% CI: 3.62–36.77) and 0.15 (95% CI: 0.06–0.36), respectively (Figure 5). A Fagan nomogram was used to illustrate the relation between PLR and NLR in Figure 6. The area under SROC (AUC) was 0.96 (95% CI: 0.94–97) (Figure 7). All data above showed a relatively high diagnostic value for miR-155 in breast cancer.

#### 3.6.2. Subgroup Analysis

A subgroup analysis of the sample (serum) analyzed was performed to investigate the potential origin of heterogeneity between studies. The pooled results of this subgroup analysis are shown in Figure 8. It can be observed that studies that analyzed microRNA-155 in serum showed a slight improvement in diagnosis accuracy: the pooled sensitivity and specificity were 88% (95% CI: 69–96%) and 94% (95% CI: 78–99%), respectively. The PLR and NLR were 15.90 (95% CI: 3.35–75.46) and 0.13 (95% CI: 0.04–0.37), respectively. The area under SROC (AUC) was 0.97 (95% CI: 0.95–98).

#### 3.6.3. Publication Bias

To distinguish the potential publication bias across the enrolled diagnostic studies, the Deeks´ funnel-plot test was performed. The Deeks’ funnel plot (Figure 9) was symmetrical and reached a *p* value of 0.31 above 0.05, indicating there is no obvious publication bias in the included studies.

## 4. Discussion

Breast cancer is the worldwide leading cancer in terms of diagnoses, with a total of 2.3 million new cases (11.7%) of breast cancer occurring yearly [1] Its mortality constitutes 6.9% of all deaths due to cancer. According to the SEER database, an authoritative source for cancer statistics in the United States, mortality from this neoplasm triples after the age of 55, from 8.3% in the 35–44 age group to 25.6% in the 55–74 age range. If breast cancer is detected when it is confined to the primary site or has disseminated to regional lymph nodes, the five-year relative survival is 93% [52]. Imaging technology has improved the diagnosis of breast cancer patients and the development of surgical techniques, while concurrent radiotherapy and chemotherapy have significantly reduced mortality compared to previous decades [53]. An ideal biomarker should have the ability to detect with high accuracy the onset of the disease even before any clinical signs appear. In addition, it should accurately discriminate between a neoplastic sample and a healthy sample, reporting variations when the neoplasm is treated or during relapses. Moreover, ideal biomarkers should have a long half-life in clinical samples and should be accessible through innocuous methods, stable, and serially reproducible with low cost [54]. MicroRNAs may provide a solution to the existing problems with current screening techniques in the early diagnosis of breast cancer. They are truly stable biomolecules, present in body fluids such as serum, plasma, or saliva. Their expression is specific and sensitive in both tissues and organs affected by a neoplasm [17,22,29,55], and their quantification method, by qRT-PCR or microarray [56], is simple and relatively inexpensive due to the worldwide adaptation that has been made to enable the extensive use of this technique for the detection of the SARS-CoV-2 viral genome. MiRNAs control the expression of target genes by either inhibiting protein translation or directly targeting the mRNA transcripts of target genes for degradation [12], with well-described roles in several cancers, including breast cancer [56]. Since miRNAs potentially have a broad influence on diverse genetic pathways, dysregulation of these small RNAs is probable to contribute to the occurrence of diseases, including cancer.

### 4.1. Size and Angiogenesis

MicroRNAs have been significantly linked to tumor development and angiogenesis. There is a balance between proangiogenic and antiangiogenic factors that work together with different expressions, releasing or reactivating factors with the ability to form new blood vessels [57]. The process of angiogenesis in normal cells is different from the angiogenic process that develops in tumor cells. Tumor blood vessels are highly disorganized, irregular in shape, and hyperpermeable, among other characteristics [58]. Tightly linked to this event is tumor size; as the tumor increases in size, aerobic conditions begin to be compromised and tissue hypoxia occurs. This hypoxia promotes the release of angiogenic factors and angiovascular mimicry, also known as EMT (endothelial-mesothelial transition). This change allows the tumor cell to restructure its phenotypic expression and show itself as an endothelial cell, and this will have very important repercussions related to its invasiveness [59].

AngiomiRNAs might be the key to this process, activating or inhibiting, both directly or indirectly, factors such as VEGF (vascular endothelial growth factor) or H1F-1 (hypoxia-inducible factor-1) [58]. Examples of anti-angiogenic miRNAs described in this systematic review include the following (alongside the targets with which they interact) miR-126 (VEGF-A) [60], miR-126-p (VEGF-A and PIK3R) [61], and miR-145 (IRS1) [62]; while among the microRNA with antagonistic roles are miR-21 (PTEN) [63], miR-93 (LATS2) [64], miR-155 (LYVE-1 and VHL) [65], and miR-373 (VEGF and cyclin) [66].

The underexpression of miRNAs related to angiogenesis inhibition and the overexpression of those related to angiogenesis promotion described in the studies included in our work support their diagnostic utility, as they are related to tumor growth, invasion, and metastasis.

### 4.2. Angiogenesis and Tumor Spreading

Neovascularization, described hereinbefore, provides a direct gateway to other organs and tissues that are colonized by tumor cells, causing tumor dissemination. When a tumor cell acquires the ability to migrate to other tissues distant from the primary cancer, usually through lymphatic or blood vessels, it can begin to reproduce and invade new organs, spreading to various areas of the body [67,68].

MicroRNAs can control invasion and metastasis processes in many ways [69], such as the cell secretion of factors such as TGF-β, involved in cell growth and the proliferation of tumor cells in early stages and contributing to EMT [70]; or through interaction with the Wnt/β-cateninn pathway, which inhibits the expression of a protein attached to the cell membrane (E-cadherin), whose loss is associated with EMT [71]. Among the microRNAs that promote these processes are miR-21 (through TGF-β and epidermal growth factor) [72], miR-10b (TGF-β1, its expression enhances E-cadherin expression) [73], miR-155 (RhoA) [74], and miR-191 (TGF-β1) [75]. Among the microRNAs that prevent metastasis are miRNA-141 and miRNA-200c [76]. In our revision, miRNAs that promote metastasis were found to be significantly overexpressed, while those that suppress it were found to be downregulated. The detection of the overexpression of miRNAs involved in angiogenesis or those that are suppressed or underexpressed for the same reason in combination with tumor markers (e.g., TGF-β, VEGF, CEA or CA15-3) could anticipate a diagnosis before tumor dissemination is produced, helping the treatment response and a reduction in sequels.

Several reports have shown that miR-155 acts as an oncomiR in human cancers by targeting tumor suppressors [77,78]. It has been revealed that miR-155 promotes cancer cell proliferation (EMT), CSC phenotype [79], chemoresistance [80,81], the evasion of apoptosis, colony formation, and tumor growth. In our meta-analysis, the AUC and corresponding 95% CI were 0.96 (0.94–97). In addition, PLR and NLR were 15.90 (95% CI: 3.35–75.46) and 0.13 (95% CI: 0.04–0.37), respectively, in serum, indicating that microRNA-155 could discriminate breast cancer patients from healthy patients with relative reliability. Traditional markers such as CEA or CA 15-3 can also benefit from microRNAs. The main disadvantage of these traditional markers is their low sensitivity, which is improved when used in combination with microRNAs, as reported by Zaleski [39] A further highlight of our work is the diagnostic value of this microRNA in different fluids. In Erbes’s research [49], miR-155 was measured in urine, and despite being underexpressed in this fluid as opposed to serum, where it is overexpressed, its sensitivity and specificity were very similar, as disclosed in our subgroup analysis, which may guide future research on other fluids with closer contact with the tumor environment, such as breast milk.

Last but not least, ethnic and racial representation is limited in the area of early detection of breast cancer using microRNAs. Studies have included only Caucasian and Asian patients. The fact that race and ethnicity correlate with breast cancer morbidity and mortality has been documented by some authors [82,83,84,85]. Generally, these characteristics are associated with socioeconomic disparities, as, in the case of black patients, several studies suggest a higher mortality and a significantly lower overall five-year survival [86,87,88]. Another fact that explains the higher mortality in ethnic minorities is the higher incidence and outcome disparities of very aggressive breast cancers, such as triple-negative breast cancer [89,90].

### 4.3. Heterogeneity: A Major Challenge to Overcome

Finally, when evaluating the results presented, some limitations must be taken into account: (1) a common feature of all of the studies reviewed both in our systematic review and in the subsequent quantitative analysis was the high biological and technical variability, which was a major source of bias. The sample preparation (the use of small, enriched RNA fraction versus total miRNA), methods used to isolate RNA (phenol/guanidinium (TRIzol, Invitrogen), miRNeasy (QIAGEN), and mirVana (ABI)…), and the differences in the selection of endogenous controls (miR-16, -39, 222-3p, among others) encourage inconsistency in the results obtained. Regarding endogenous controls, miR-16 is recognized by several authors as the most stably expressed microRNA in breast cancer patients and healthy controls [91,92,93]. (2) Racial factors were not represented in their scope. The included articles presented a monotonous Caucasian and Asian population. The African race should be studied more widely, especially because of the greater incidence of breast cancer in this community. Therefore, more researchers should pay attention to the impact of racial factors in subsequent studies. (3) Our work only included articles published in English, but did not cover articles in other languages. (4) The sample size was too small, which may undermine the reliability of our results. Therefore, more well-designed studies based on larger samples and sufficient data are required to verify the diagnostic value of miR-155. (5) Subgroup analysis could not be performed in our meta-analysis because of the limited data for adjustment for covariates such as TNM stage, histologic type, assay, and so on [94].

## 5. Conclusions

In summary, the results reported in the present study may be useful for future confirmatory analyses more focused on the expression of preselected miRNAs in breast cancer patients. The miRNAs constitute a new avenue to explore to complement the use of classical breast cancer biomarkers, thereby improving sensitivity and specificity, and to be used as an isolated approach, particularly when these are pooled to form panels. Based on the results of our meta-analysis, miR-155 might be a promising diagnostic biomarker for this patient population. It is a blood-based biomarker that is easy to obtain and non-invasive. At the same time, this biomarker is highly stable. In the same way that certain miRNA expression profiles in blood appear to be specific—such as miR-122, expressed preferentially in liver and miR-133, expressed in muscle—miR-155 could be breast-specific. Moreover, it is noteworthy that different laboratories studying the same disease find the same circulating miRNAs as potential biomarkers, which strengthens the idea that miRNAs may be valid for diagnosing cancer and other diseases. Nevertheless, large-scale, well-designed, multicenter studies should be conducted to clarify the mechanism of miR-155 overexpression in breast cancer.

## Figures and Tables

**Figure 1 ijms-24-08270-f001:**
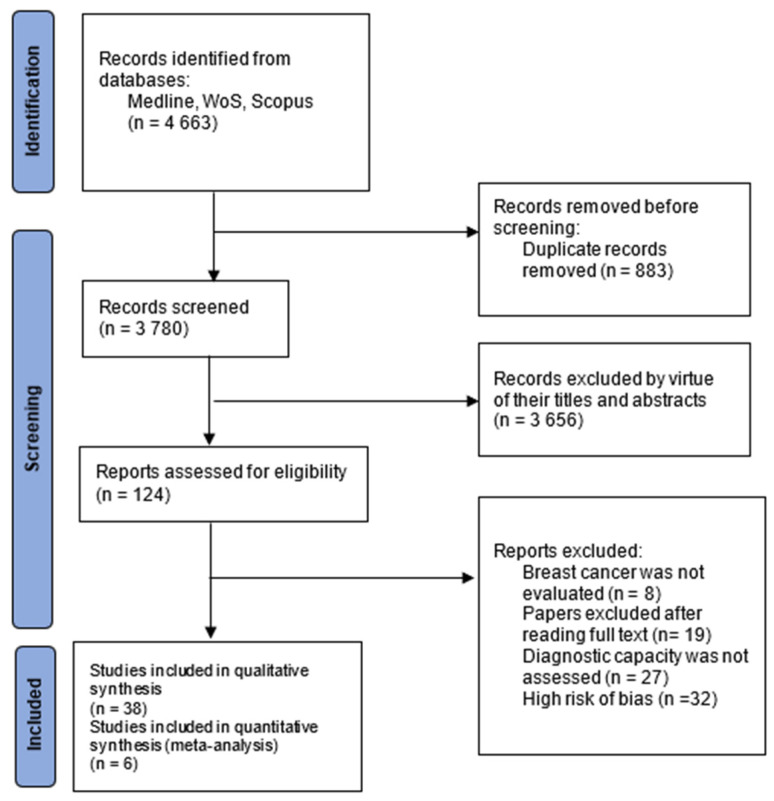
Adapted PRISMA flow diagram of trials.

**Figure 2 ijms-24-08270-f002:**
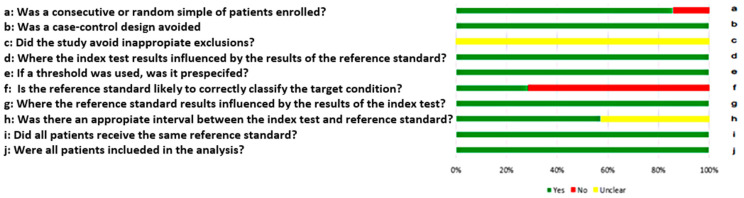
QUADAS-2 diagram of studies selected for the meta-analysis.

**Figure 3 ijms-24-08270-f003:**
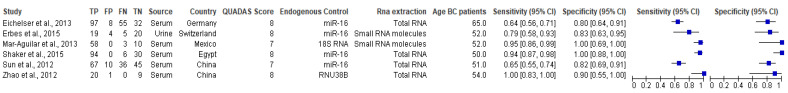
Summary of studies [18,19,24,37,47,49] included in miR-155 meta-analysis.

**Figure 4 ijms-24-08270-f004:**
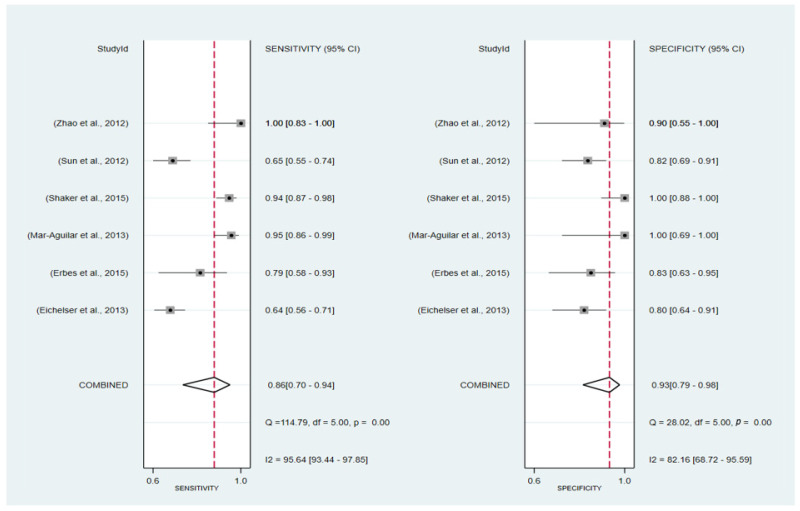
Forest plot of the overall pooled sensitivity and specificity. Studies [18,19,24,37,47,49].

**Figure 5 ijms-24-08270-f005:**
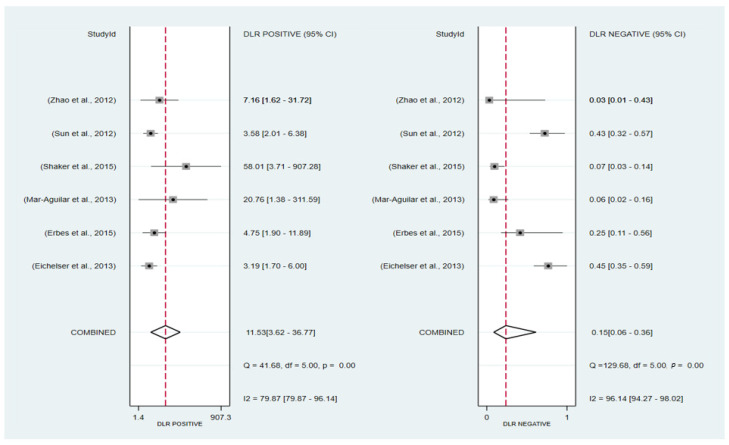
Forest plot of the overall pooled positive and negative likelihood ratios. Studies [18,19,24,37,47,49].

**Figure 6 ijms-24-08270-f006:**
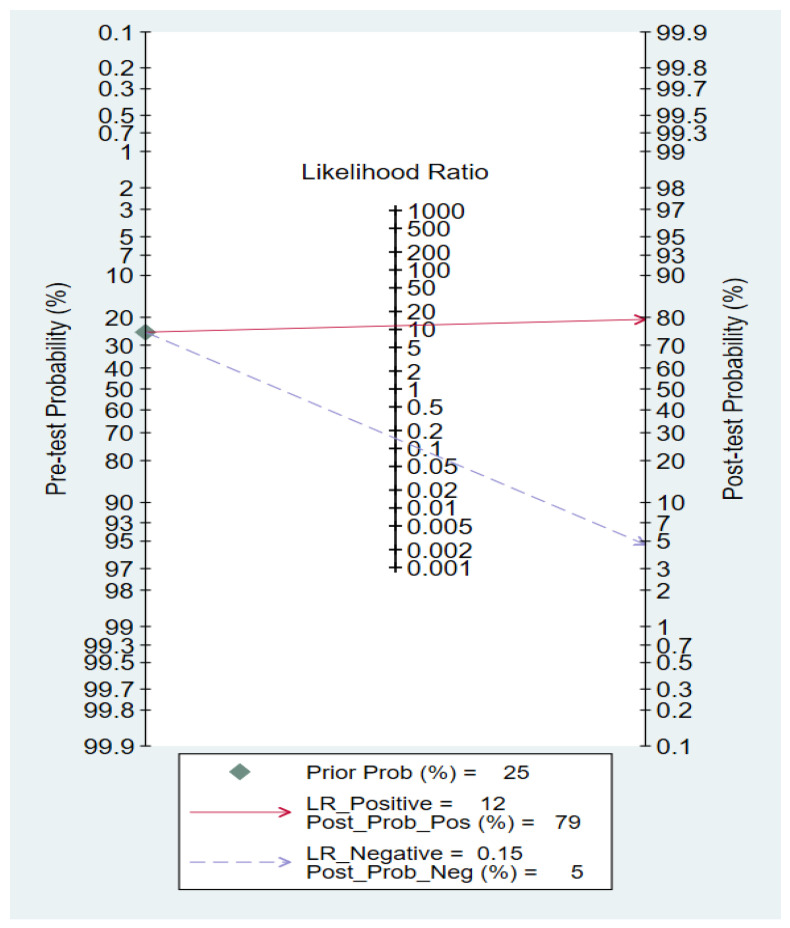
Fagan nomogram of microRNA-155 for diagnosis breast cancer.

**Figure 7 ijms-24-08270-f007:**
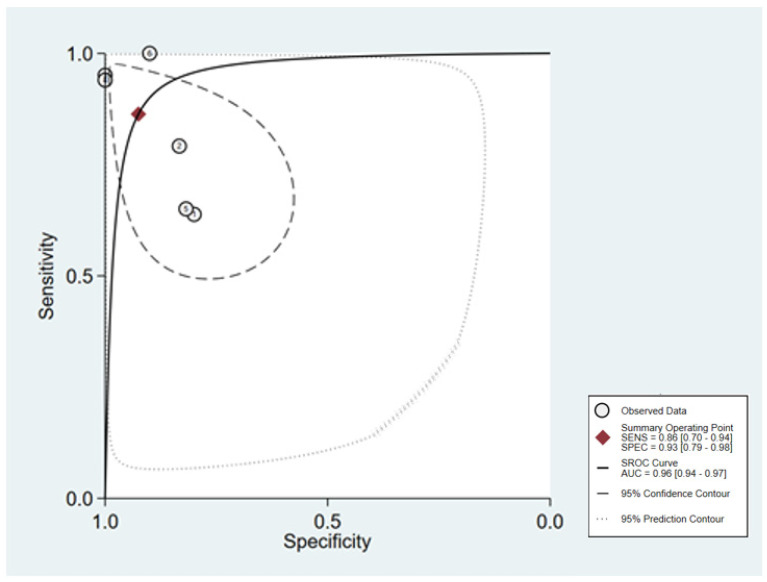
Summary receiver operating curve (SROC) for breast cancer diagnosis.

**Figure 8 ijms-24-08270-f008:**
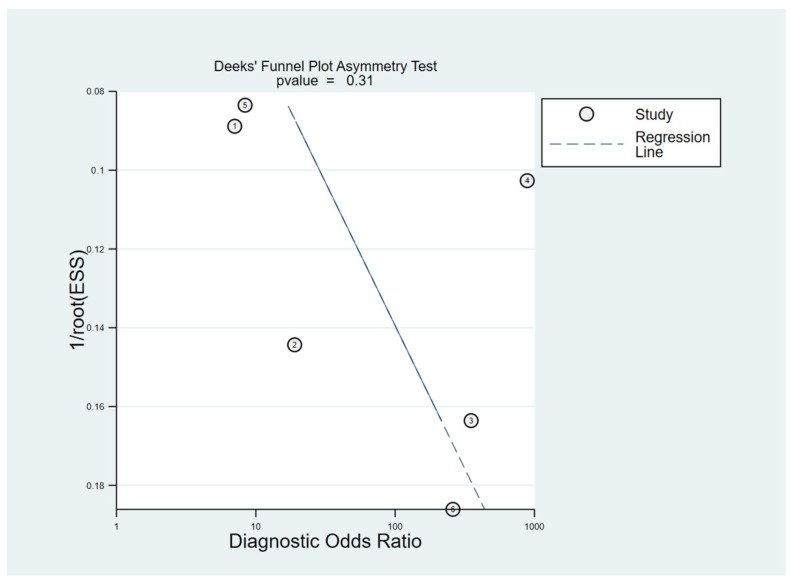
Pooled results of the sample to investigate the potential origin of heterogeneity, presented in a Deeks’ funnel plot.

**Figure 9 ijms-24-08270-f009:**
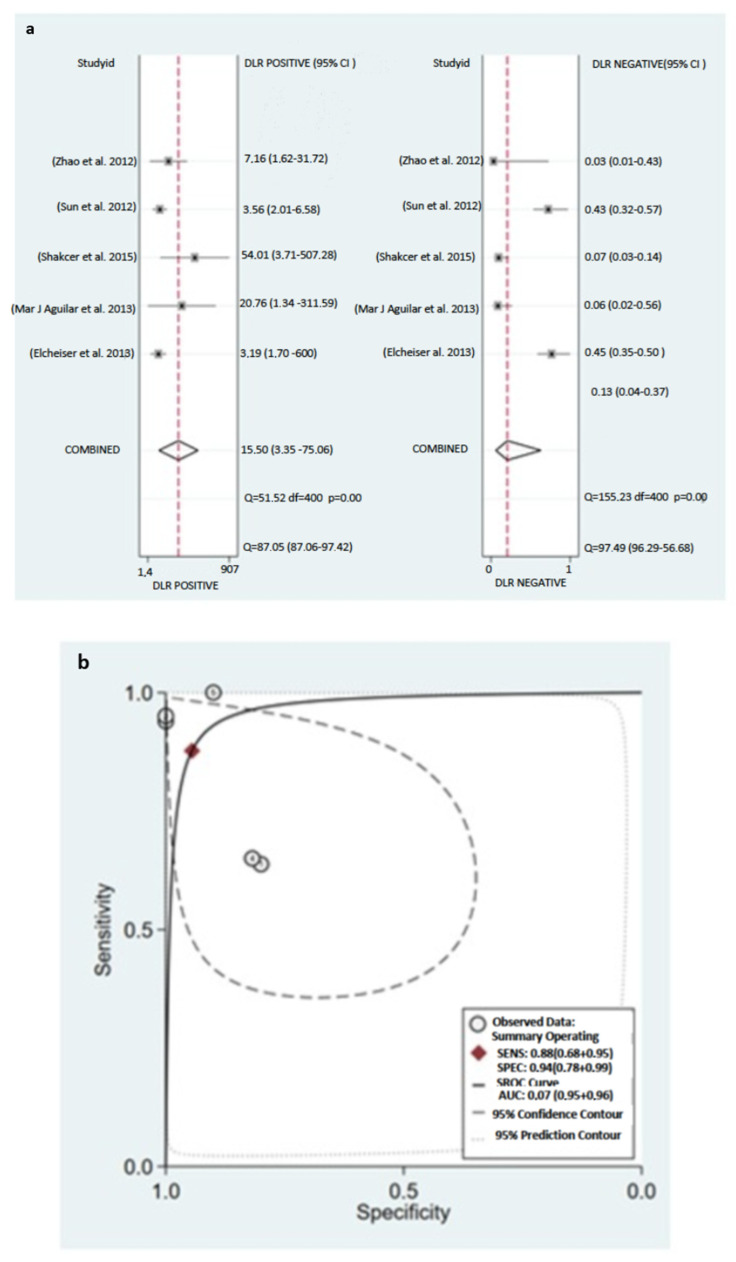
Subgroup analysis based on the source (serum) type of (**a**) sensitivity and specificity, (**b**) SROC curve, and (**c**) positive and negative likelihood ratios. Studies [18,19,24,37,47,49].

**Table 1 ijms-24-08270-t001:** Studies comparing the expression level of miRNAs in cases and controls.

miRNA/s	References	Normalization	Sample	n	Cases	Controls	Conclusions
**miR-21**	[15]	miR-16	Serum	122	102	20	miRNA-21 concentrations differed between cases and controls and enabled the differentiation of localized disease from metastases.
**miR-21**	[16]	miR-16	Serum	89	50	39	The mean serum levels of miR-21 were higher in different types of cancer compared to control (including breast cancer). The expression of miR-21 did not correlate to other aspects of breast cancer such as estrogen receptor, progesterone receptor, menopause status, and KI-67. Similarly, miR-21 expression was not associated with metastasis status. There were nine males in the control group.
**miR-99a-5p**	[17]	RNU38B	Serum and tissue	203	105	98	The data reflected a significant underexpression of miR-99a-5p in breast cancer tissue versus healthy tissue. However, serum levels reflected overexpression in cases versus controls. Only the testing cohort was considered.
**miR-155**	[18]	miR-39	Serum	158	103	55	MiR-155 levels were significantly elevated in the case group compared to control. The ROC curve = 0.801 with a specificity of 81.8%, suggesting that the expression of miR-155 allows one to discriminate the cases from the controls, and its expression decreases with a good response to treatment.
**miR-155**	[19]	RNU38B	Serum	30	20	10	miR-155 levels were significantly elevated in the cases group.
**miR-155**	[20]	miR-16	Serum	117	102	15	The expression of miR-155 was significantly higher in the cases group. After receiving treatment (surgery and chemotherapy), the expression decreased significantly. miR-155 overexpression was also related to tumor size and stage.
**miR-202**	[21]	miR-16	Serum	60	30	30	miR-202 was found to be significantly overexpressed in the cases. Sensitivity and specificity were 90 and 93%, respectively. The positivity was 100% for Stage I, 90% for Stage II, and 80% for Stage III, demonstrating great utility for the diagnosis of BC in early stages, in addition to showing predictive value (risk of 9.6 times higher, *p* < 0.001)
**miR-1204**	[22]	GAPDH	Serum and tissue	182	144	38	The expression of miR-1204 was significantly higher both in the tissue and in the serum of the cases compared to the controls, allowing the cases to be discriminated from the controls with a ROC curve of 0.854.
**miR-34a** **miR-10b**	[23]	miR-16	Serum	118	89	29	The microRNAs analyzed were significantly elevated in the cases. miR-10b and miR-34a were significantly higher in patients with metastases compared to patients with localized breast cancer. Elevation of miR-34a correlated with advanced stages.
**miR-10b** **miR-21** **miR-145** **miR-155** **miR-191** **miR-382**	[24]	18S RNA	Serum	71	61	10	The concentrations of all miRNAs were significantly higher in cases. The ROC curve showed that three of them (miR-145, miR-155, and miR-382) are potential biomarkers. Possible differences in the expression of the cases were also analyzed, although they were not significant (different stages)
**miR-21** **miR-106a** **miR-126** **miR-155** **miR-199a** **miR-335**	[25]	miR-16	Serum	108	68	40	The concentrations of the microRNAs were significantly different between the cases and the controls. miR-21, miR-106a, and miR-155 were overexpressed and miR-126, miR-199a, and miR-335 were underexpressed.
**miR-16** **miR-21** **miR-145** **miR-451**	[26]	RNU6B	Serum	270	170	100	miR-21 and miR-451 were found to be significantly overexpressed in the cases, while miR-145 was underexpressed with respect to the controls. The combination of miR-145 and miR-451 presented a ROC curve of 96%, with an optimal sensitivity of 90% and specificity of 96%. These markers are not elevated for other types of cancer and are useful in different stages of breast cancer, with higher diagnostic values in some types of breast neoplasms, such as DCI (ductal carcinoma in situ), which obtained a ROC curve of 98%
**miR-21** **miR-146a**	[27]	miR-16	Serum	22	14	8	The expression of miR-21 and miR-146a were found to be significantly higher in the cases compared to the controls.
**miR-21** **miR-92a**	[28]	RNU6	Serum	120	100	20	Significant overexpression of miR-21 and significant underexpression of miR-92a. Subsequent analysis showed that low miR-92a and high miR-21 levels were associated with tumor size and lymph node staging.
**miR-21** **miR-1246**	[29]	miR-54	Serum and tissue	32	16	16	Several microRNAs were encapsulated or highly enriched and selectively secreted by tumor exosomes. Both miR-21 and miR-1246 showed significantly high expression in the cases. miR-122 and let-7a are also considered to be potential biomarkers as they are selectively secreted.
**let-7a** **miR-10b** **miR-16** **miR-21** **miR-145** **miR-155** **miR-195**	[30]	miR-16	Serum	146	83	63	Only miR-195 was found to be significantly overexpressed in patients with breast cancer specifically, compared to cases of other neoplasms and healthy controls. Furthermore, a positive correlation between miR-195 levels and tumour size was demonstrated. On the other hand, let-7a was significantly elevated in the cases (with the exception of malignant melanoma). In patients with breast neoplasms, the simultaneous use of three miRNAs (let-7a, miR-195, and miR-155) led to a sensitivity of 94%.
**miR-99a** **miR-145** **miR-151-3p** **miR-205** **miR-210** **miR-361-5p**	[31]	miR-16	Serum	28	21	7	miR-145 was found to be underexpressed in the cases compared to the controls. miR-210 was found to be overexpressed in the cases. Underexpression of miR-99 and overexpression of miR-155-3p. In invasive breast cancer, the underexpression of miR-145 and overexpression of miR-210 were observed. In metastatic breast cancer, the underexpression of miR-205 and overexpression of miR-361-5p were observed.
**miR-1246** **miR-1307-3p** **miR-4634** **miR-6861-5p** **miR-6875-5p**	[32]	miR-149-3p	Serum	4116	1280	2836	A combination of five miRNAs was able to discriminate cases from controls with 97.3% sensitivity, 82.9% specificity, and 89.7% precision for the breast cancer cohort. In addition, for early-stage breast cancer, the sensitivity was 98%.
**miR-133a** **miR-155** **p53** **CEA** **CA-15.3**	[33]	SNORD68	Serum	80	60	20	While miR-155 was significantly overexpressed in the cases, miR-133a was significantly underexpressed. Both CEA and CA-15.3 were found to be significantly elevated in the serum of the cases. A possible relationship was found between miR-133a and tumour stage (underexpression—higher stage) and between miR-155 and lymph node metastasis (overexpression—lymph nodes involved).
**miR-21** **MMP-1**	[34]	miR-16	Urine	48	22	26	miR-21 was found to be significantly underexpressed in the cases, in contrast to the matrix metalloproteiase-1 (MMP-1), which was found to be significantly overexpressed. Combined, sensitivity reached 95% and specificity was 79%.
**Let-7b-5p** **miR-122-5p** **miR-146b-5p** **miR-210-3p, 215-5p**	[35]	miR-39	Serum	514	257	257	Eleven miRNAs were significantly deregulated. These were analyzed in seventy-two samples and only five were found to be consistently overexpressed in the cases. The diagnostic capacity of five miRNAs yielded a ROC curve of 0.978, 94% sensitivity, and 88.9% specificity.
**miR-21** **miR-155** **let-7c** **PTEN**	[36]	RNU44	Serum	93	45	48	The expression of miR-21 was significantly higher in the cases compared to the controls. On the contrary, miR-155, let-7c, and PTEN were found to be significantly underexpressed in the cases.
**miR-29b-2** **miR-155** **miR-197** **miR-205**	[37]	miR-16	Serum	130	100	30	The expression of miR-155 was significantly higher in the cases versus the controls, while miR-205 was significantly underexpressed.
**miR-16** **miR-21** **miR-29c** **miR-145** **miR-191** **miR-210** **miR-222**	[38]	miR-222-3p	Serum	68	35	33	Only three miRNAs were significantly overexpressed in the cases (miR-145, -191, and -21). Individually, the ROC curves of miR-145 and -191 were 0.931 and 0.904, respectively, and in combination, this rose to 0.984, allowing for the accurate differentiation of cases versus controls.
**Let-7c** **miR-21** **miR-34a** **miR-92a** **miR-155** **miR-222** **CEA** **CA 15-3** **CA 125**	[39]	miR-16	Serum	83	55	28	The cases were shown to have a higher expression of miRNAs. The expression of miR-34a was significantly lower in the cases, unlike CEA and CA 15-3. The combination of CEA or CA 15-3 together with miR-34a yielded ROC curves of 0.844 and 0.800, respectively. The specificity of the combination of miR-34a and CA 15-3 was 95%. When the cohort of patients with breast cancer was compared with benign diseases, both sensitivity and specificity and the ROC curve showed better results.
**miR-23a** **miR-29b** **miR-130** **miR-144** **miR-148a** **miR-152** **miR-182**	[40]	U6	Serum and tissue biopsy	202	106	96	The expressions of miR-23a-3p, -130a-5p, -144-3p, -148a-3p, and 152-3p were lower in the plasma of the cases compared to the controls. miR-130a-5p, -144-3p, and 152-3p were also found to be underexpressed in the tissue of the cases. The expression of miR-23a-3p, -144-3p, and 152-3p was related to ER and PR + status, in addition to showing significant differences in stage, especially in the early stages.
**miR-16** **miR-25** **miR-222** **miR-324-3p**	[41]	miR-39	Serum	152	76	76	Four miRNAs were found to be significantly overexpressed in cases versus controls. The AUC was 0.928 for the miRNA profile, with a sensitivity and specificity of 0.921 and 0.934, respectively. Validation of the cohort was considered.
**miR-21** **miR-24** **miR-202** **miR-206** **miR-219B** **miR-223** **miR-373** **miR-1246** **miR-6875**	[42]	miR-16	Serum	136	80	56	The combination of two or more miRNAs of the proposed profile reported more precise results than the use of a single miRNA individually. The combination with the highest precision was that of miR-1246, -206, -24, and -373, obtaining a 98% sensitivity, 96% specificity and 97% precision. Combinations such as that made up of miR-1246, -206 and -24 achieved 100% sensitivity, 93% specificity, and 97% precision. Validation of the cohort was considered.
**miR-15a** **miR-18a** **miR-107** **miR-133a** **miR-139-5p** **miR-143** **miR-145** **miR-365** **miR-425**	[43]	miR-10b miR-30a	Serum	72	48	24	Nine miRNAs were found to be deregulated in the cases versus the controls. Subsequently, a panel composed of nine miRNAs validated in 111 serum samples was developed, yielding an AUC of 0.665. The authors also reflected the possibility of analyzing the risk of suffering from breast cancer ER + with this miRNA profile.
**miR-155**	[44]	RNU6B	Serum	50	30	20	miR-155 was found to be significantly overexpressed in the cases versus the controls (*p* < 0.001). In the cases, the expression of this miRNA varied significantly depending on the tumor stage.

**Table 2 ijms-24-08270-t002:** Summary of dysregulated miRNAs.

miRNA	References	Dysregulation	Sample Size (Total)	Cases (Total)	Control (Total)	Relative Use
Let-7a	[30,45]	Upregulated	338	231	107	Diagnosis and prognosis
miR-10b	[23,24,30,46,47]	No significative	359	270	89	Diagnosis and prognosis
miR-16	[26,30,38,41,46]	No significative	536	253	183	Normalization
miR-19a	[48]	Upregulated	84	63	21	Diagnosis and prognosis
miR-21	[15,16,24,25,26,27,28,29,30,34,36,38,39,42,49,50,51]	Upregulated Downregulated (Urine [49])	1791	1083	708	Diagnosis and prognosis
miR-24	[42,48]	Upregulated	84	63	21	Diagnosis and prognosis
miR-34a	[23,39,46,47]	Upregulated Downregulated [39,46]	563	406	157	Diagnosis and prognosis
miR-92a	[28,39]	Downregulated	251	203	48	Diagnosis
miR-93	[47]	Upregulated	192	152	40	Diagnosis (BM TN)
miR-99a-5p	[17]	Upregulated (Serum) Downregulated (Tissue biopsy)	174	89	75	Diagnosis
miR-106a	[25]	Upregulated	108	68	40	Diagnosis
miR-125b	[24,49]	Upregulated Downregulated (urine [49])	119	85	34	Diagnosis
miR-126	[25]	Downregulated	108	68	40	Diagnosis
miR-133a	[33,43]	Downregulated	191	120	71	Diagnosis and prognosis
miR-141	[23]	Upregulated	118	89	29	Diagnosis
miR-145	[24,26,30,31,38,43]	Downregulated	644	380	264	Diagnosis
miR-146a	[27]	Upregulated	22	14	8	Diagnosis
miR-155	[18,19,20,23,24,25,30,33,36,37,39,44,46,47,48,49]	Upregulated	1637	1154	483	Diagnosis and prognosis
miR-199a	[25]	Downregulated	108	68	40	Diagnosis
miR-181b	[46,48]	Upregulated	254	183	71	Diagnosis and prognosis
miR-191	[24,38]	Upregulated	139	96	43	Diagnosis
miR-195	[30,45]	Upregulated	338	231	107	Diagnosis and prognosis
miR-199a	[25]	Downregulated	108	68	40	Diagnosis
miR-202	[21]	Upregulated	60	30	30	Diagnosis and predictor
miR-205	[31,37]	Upregulated	218	151	67	Diagnosis and prognosis
miR-210	[31,35]	Upregulated	542	278	264	Diagnosis
miR-222	[39,41]	Upregulated	235	131	104	Diagnosis
miR-335	[25]	Downregulated	108	68	40	Diagnosis
miR-373	[42,47]	Upregulated	328	232	96	Diagnosis (HER2+)
miR-451	[26,49]	Upregulated Downregulated (Urine [49])	268	144	124	Diagnosis
miR-1204	[32]	Upregulated	182	144	38	Diagnosis
miR-1307	[32]	Upregulated	4116	1280	2836	Diagnosis
miR-1246	[29,32,42]	Upregulated	4284	1376	1308	Diagnosis
miR-4634	[32]	Upregulated	4116	1280	2836	Diagnosis
miR-6861-5p	[32]	Upregulated	4116	1280	2836	Diagnosis
miR-6875-5p	[32,42]	Upregulated	4252	1360	2892	Diagnosis

## Data Availability

All data relating to this study are available in the custody of the corresponding author.

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
