# Peer review of "MicroRNA Dysregulation in Early Breast Cancer Diagnosis: A Systematic Review and Meta-Analysis"

_ijms, 2023, doi:10.3390/ijms24098270_

Round 1

Reviewer 1 Report

Carrido-Palacios et al. made quite important discussion based on meta-analysis and found that miR-155 might be a promising diagnostic biomarker in this systematic review. Therefore, this article is suitable to publish in this special issue.

Minor comments;

1. The title of Figure 7 should explain more detail like the others.

2. Page 6, line 275: "AngiomiRs" may be better to change to "AngiomiRNAs" because unite the style with "oncomiRNAs".

Author Response

R1

Comments and Suggestions for Authors

Carrido-Palacios et al. made quite important discussion based on meta-analysis and found that miR-155 might be a promising diagnostic biomarker in this systematic review. Therefore, this article is suitable to publish in this special issue.

Thank you very much for your comments.

Minor comments;

  1. The title of Figure 7 should explain more detail like the others.

The title of figure 7 is revised.

  1. Page 6, line 275: "AngiomiRs" may be better to change to "AngiomiRNAs" because unite the style with "oncomiRNAs".

The requested revision is made.

Thank you very much again

Reviewer 2 Report

In this paper, the authors present a meta-analysis of articles aimed at investigating possible correlations between microRNA expression levels and diagnosis in human breast cancer.

The authors ask themselves the question of identifying, starting from data published in the literature, a set of microRNAs that may be of interest in the prediction of the onset of breast cancer. The paper is grounded on the PROSPERO protocol “CRD42021256338”, originally entitled “Breast milk microRNAs for screening breast cancer: sistematic review and meta-analysis”.

Basically, within this aim, the authors investigated current literature focusing on observational case-control based studies analyzing the PCR expression level of  microRNAs which had been published and published within the last 10 years were selected, reporting comparison between women with breast cancer and healthy controls. As a major result, based on the results of the proposed meta-analysis, miR-155 might be a promising diagnostic biomarker.

The article is a meta-analysis of published papers, it does not present new experimental data and/or new bioinformatic analyses.

My opinion is that basically this article is properly and correctly written and designed, but that the overall validity and importance of the results is unclear. Also, some molecular biology terms need some clarification.

Major:

The fundamental point of this meta-analysis is that, if I have understood correctly, the overall number of articles taken into consideration at the end of the selection process is very limited, as the final analysis is essentially concentrated on 6 papers, shown in table 3.

Given the vast number of articles present in the field of molecular biology on microRNA in cancer and in particular breast cancer, it is honestly not very clear to me whether the main result reported in the current version of the manuscript (miR-155) is essentially due to the very small database used in this study or this represent a real data somewhat useful. In other words, I do not believe that the analyzes reported in this article truly represent an advance in the field. 

Furthermore, the authors here mainly use microRNA expression data from PCR experiments. A large body of literature on breast cancer is available considering microRNA-seq data. It is my opinion that an updated review (year 2023) must also consider this kind of information.

Minor:

Some sentences contain molecular biology statements wrong or partially confounding, e.g:

-       “miRNAs concentrations” - I think is misleading to use the word "concentrations"; are the authors meaning “expression level”?

-       “These molecules represent only 2-3% of the human genome” - this is basically wrong or unclear. The human genome is 3 x 10^9 base pairs long. On the human genome we have currently around 2.000 miRNAs annotated, but each (mature) miRNA is just 20-25 base pairs long, thus representing a tiny fraction of the human genome, in terms of sequence length.

Author Response

R2

Comments and Suggestions for Authors

In this paper, the authors present a meta-analysis of articles aimed at investigating possible correlations between microRNA expression levels and diagnosis in human breast cancer.

 The authors ask themselves the question of identifying, starting from data published in the literature, a set of microRNAs that may be of interest in the prediction of the onset of breast cancer. The paper is grounded on the PROSPERO protocol “CRD42021256338”, originally entitled “Breast milk microRNAs for screening breast cancer: sistematic review and meta-analysis”.

 Basically, within this aim, the authors investigated current literature focusing on observational case-control based studies analyzing the PCR expression level of  microRNAs which had been published and published within the last 10 years were selected, reporting comparison between women with breast cancer and healthy controls. As a major result, based on the results of the proposed meta-analysis, miR-155 might be a promising diagnostic biomarker.

 The article is a meta-analysis of published papers, it does not present new experimental data and/or new bioinformatic analyses.

 My opinion is that basically this article is properly and correctly written and designed, but that the overall validity and importance of the results is unclear. Also, some molecular biology terms need some clarification.

Thank you very much for your comments

Major:

 The fundamental point of this meta-analysis is that, if I have understood correctly, the overall number of articles taken into consideration at the end of the selection process is very limited, as the final analysis is essentially concentrated on 6 papers, shown in table 3.Given the vast number of articles present in the field of molecular biology on microRNA in cancer and in particular breast cancer, it is honestly not very clear to me whether the main result reported in the current version of the manuscript (miR-155) is essentially due to the very small database used in this study or this represent a real data somewhat useful. In other words, I do not believe that the analyzes reported in this article truly represent an advance in the field. Furthermore, the authors here mainly use microRNA expression data from PCR experiments. A large body of literature on breast cancer is available considering microRNA-seq data. It is my opinion that an updated review (year 2023) must also consider this kind of information.

For the development of the meta-analysis only six articles were chosen as the rest of the articles did not pass the minimum quality criteria (to evaluate this area we used the quality assessment tool recommended for diagnostic accuracy meta-analyses QUADAS Score), did not provide sufficient data or the authors did not respond to our requests for information as it is necessary to know not only the sample, but also the exact distribution of true positives, true negatives, false positives and false negatives.

 Minor:

Some sentences contain molecular biology statements wrong or partially confounding, e.g:

-       “miRNAs concentrations” - I think is misleading to use the word "concentrations"; are the authors meaning “expression level”?

 The requested data is reviewed

-       “These molecules represent only 2-3% of the human genome” - this is basically wrong or unclear. The human genome is 3 x 10^9 base pairs long. On the human genome we have currently around 2.000 miRNAs annotated, but each (mature) miRNA is just 20-25 base pairs long, thus representing a tiny fraction of the human genome, in terms of sequence length.

 The requested data is reviewed

Thank you very much again

Reviewer 3 Report

In the manuscript the authors conduct a wide systematic review and meta analysis on miRNA in breast cancer to evaluate their potential accuracy as diagnostic biomarkers, identifying in miR-155 a promising candidate.

The manuscript is well written and data retrieval and analysis are performed in correct and accurate manner. Conclusions are appropriate according to the results obtained, while limitations of the study are adequately discussed.

My only minor points are for the introduction section.

-          The manuscript is centred exclusively on circulating miRNA . Then, it should be added a brief paragraph  better explaining the focus on circulating miRNA and generally, the potential value/reliability of miRNAs as circulating biomarker

-          The table 1 related to the oncogenic miRNAs should be removed, it could be easily replaced by a reference. The oncogenic role of miRNAs is well ascertained, furthermore both circulating and tissue miRNAs are listed, so there is no specific added value for it.

Author Response

R3

Comments and Suggestions for Authors

In the manuscript the authors conduct a wide systematic review and meta analysis on miRNA in breast cancer to evaluate their potential accuracy as diagnostic biomarkers, identifying in miR-155 a promising candidate.

The manuscript is well written and data retrieval and analysis are performed in correct and accurate manner. Conclusions are appropriate according to the results obtained, while limitations of the study are adequately discussed.

Thank you very much for your comments

My only minor points are for the introduction section.

-          The manuscript is centred exclusively on circulating miRNA . Then, it should be added a brief paragraph  better explaining the focus on circulating miRNA and generally, the potential value/reliability of miRNAs as circulating biomarker

The requested paragraph is added.

-          The table 1 related to the oncogenic miRNAs should be removed, it could be easily replaced by a reference. The oncogenic role of miRNAs is well ascertained, furthermore both circulating and tissue miRNAs are listed, so there is no specific added value for it.

 We do not clearly understand what is requested here: delete table 1 by a reference? table 1 alone there are more than 15 references. The team of authors of this article understands that this table adds a lot of information to the article. At the same time we believe that we must be in error and really what you are asking for is another aspect.  Please, if you need another revision, please specify this request more clearly.

Thank you very much again